# Doxycycline induces Hok toxin killing in host *E. coli*

Chinwe Uzoma Chukwudi[1,2]*, Liam Good[2]

1 Department of Veterinary Pathology and Microbiology, University of Nigeria, Nsukka, Nigeria,
2 Department of Pathology and Infectious Diseases, Royal Veterinary College, University of London, London, England, United Kingdom

* chinwe.chukwudi@unn.edu.ng

**Data Availability Statement:** All relevant data are within the paper.

**Funding:** CC received Commonwealth Academic Staff Scholarship and Fellowship from the Commonwealth Scholarship Commission in the UK (NGCF-2017-199) for the period of this work.

## Abstract

The antibacterial efficacy of the tetracycline antibiotics has been greatly reduced by the development of resistance, hence a decline in their clinical use. The *hok/sok* locus is a type I toxin/antitoxin plasmid stability element, often associated with multi-drug resistance plasmids, especially ESBL-encoding plasmids. It enhances host cell survivability and pathogenicity in stressful growth conditions, and increases bacterial tolerance to β-lactam antibiotics. The *hok/sok* locus forms dsRNA by RNA:RNA interactions between the toxin encoding mRNA and antitoxin non-coding RNA, and doxycycline has been reported to bind dsRNA structures and inhibit their cleavage/processing by the dsRNase, RNase III. This study investigated the antibacterial activities of doxycycline in hok/sok host bacteria cells, the effects on *hok/sok*-induced changes in growth and the mechanism(s) involved. Diverse strains of *E. coli* were transformed with *hok/sok* plasmids and assessed for doxycycline susceptibility and growth changes. The results show that the *hok/sok* locus increases bacterial susceptibility to doxycycline, which is more apparent in strains with more pronounced *hok/sok*-induced growth effects. The increased doxycycline susceptibility occurs despite β-lactam resistance imparted by *hok/sok*. Doxycycline was found to induce bacterial death in a manner phenotypically characteristic of Hok toxin expression, suggesting that it inhibits the toxin/antitoxin dsRNA degradation, leading to Hok toxin expression and cell death. In this way, doxycycline could counteract the multi-drug resistance plasmid maintenance/propagation, persistence and pathogenicity mechanisms associated with the *hok/sok* locus, which could potentially help in efforts to mitigate the rise of antimicrobial resistance.

## Introduction

Many cellular RNAs play a regulatory role, and the secondary structures that are important to their biological activities are formed through folding and formation of double-stranded regions (dsRNA). Antisense RNAs are examples of regulatory RNAs and often act by down-regulating gene expression post-transcriptionally following the formation of dsRNA with their target messenger (sense) RNAs. By forming a duplex with the mRNA of the regulated gene via base-pairing interactions, the dsRNA regions may block ribosome recognition of the mRNA

**Competing interests:** The authors have declared that no competing interests exist.

or lead to recognition and cleavage by double-stranded ribonucleases [1, 2]. In bacteria, many antisense RNAs are encoded on plasmids, and are often involved in regulation of plasmid replication and copy number control [3]. Moreover, many plasmid and chromosomally-encoded antisense RNAs in bacteria are known to act as type I antitoxins by down-regulating the expression of toxic proteins [4–6]. In addition, some antisense RNA antitoxins regulate the expression of toxins associated with stress response [4].

A well-established example of a *cis*-encoded regulatory antisense RNA in bacteria is the Sok (suppression <u>of</u> <u>k</u>illing) antitoxin commonly found in *E. coli*. The Sok antisense RNA regulates the expression of the *hok* (<u>host</u> <u>k</u>illing) mRNA by binding to the *hok* mRNA and initiating RNase III decay of the duplex, thereby inhibiting the translation of the *hok* transcript [7, 8]. Sok RNA is more labile (half-life of about 30s) than the stable *hok* mRNA (half-life of hours), which persists for much longer within new daughter cells following cell division [9–11]. In daughter cells containing the plasmid, the continued rapid transcription of Sok antisense RNA from its strong promoter ensures inhibition of *hok* expression. In cells that lose the plasmid during division, the acquired Sok RNA are more rapidly degraded than the *hok* mRNA, thus releasing the stable *hok* mRNA for translation, leading to subsequent cell death by the toxin produced (Fig 1). The toxin causes damage to the cell membrane, which appears as a characteristic morphology referred to as 'ghost cell' [11–14]. This ensures that all surviving daughter cells inherit the plasmid, as those that do not inherit the plasmid during replication are quickly killed by the Hok toxin [15]. Additionally, the *hok/sok* locus has been found to enhance bacterial stress response and improve growth in growth-limiting conditions such as high temperature, low cell density and antibiotic treatment [16].The R1 plasmid in which the *hok/sok* locus was originally discovered carries several genes that encode resistance traits, and in this way is able to impart multi-drug antibiotic resistance (ampicillin, chloramphenicol, kanamycin, streptomycin and sulphonamides) to its host bacteria [17–19]. Subsequently, the *hok/sok* locus has been found in many plasmids that encode extended spectrum beta-lactamases, especially CTX-M ESBLs [20], as well as in the chromosomes of some enterobacteria [21]. In *E. coli*, the chromosomally-encoded *hok/sok* loci are more abundant in pathogenic strains [22]. In laboratory strains (K-12), certain chromosomal loci appear to be inactivated by insertion elements located close to the toxin reading frames, which are absent in wild type cells [4]. Unlike the plasmid-encoded locus, the chromosomally-encoded *hok/sok* loci do not mediate plasmid stabilization, and their *hok* toxin mRNAs are poorly translated *in vitro* [21]. Hence, translational activation may require induction by unknown signals in living cells.

Doxycycline has been shown to bind dsRNAs and inhibit cleavage by RNase III [23]. It has also been shown to inhibit pre-rRNA processing and the formation of mature rRNA in *E. coli*; a pathway primarily dependent on RNase III cleavage [24]. Here, we hypothesize that doxycycline could inhibit RNase III degradation of the *hok/sok* toxin/antitoxin dsRNA complex, leading to the release of the *hok* mRNA for translation as the labile Sok RNA disintegrates and consequent cell death. Given that the *hok/sok* locus is often associated with antibiotic resistance genes and propagation mechanism, this study investigated the effect of doxycycline on *hok/sok* host bacteria survival and associated antibiotic resistance traits.

## Materials and methods

### Strains and plasmids

*E. coli* strains and plasmids used in this work are listed in Table 1. Cells were grown on LB agar/broth, with the addition of 100 μg/ml of ampicillin (Amp) or 30 μg/ml of

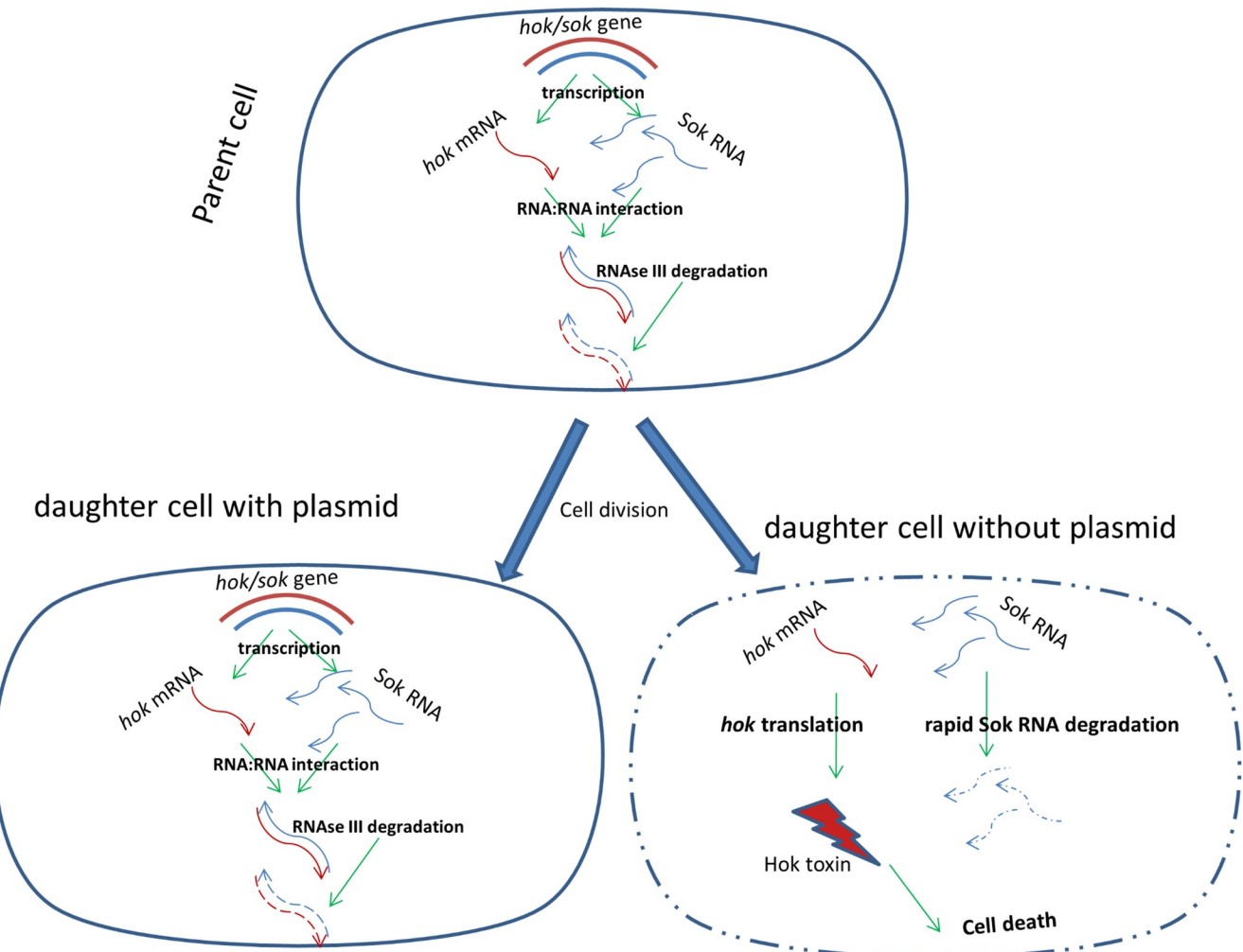

**Fig 1. Model of the *hok/sok* RNA:RNA interactions involved in plasmid stabilization *via* post-segregational killing.** In cells containing the *hok/sok* plasmid, Sok (suppression of killing) antitoxin RNA forms dsRNA with the *hok* (host killing) toxin mRNA which are degraded by RNase III to prevent the expression of the toxin. In the case of plasmid loss (e.g. following cell division), Sok RNA is rapidly degraded, releasing the *hok* mRNA for translation. The Hok toxin produced results in a 'ghost cell' morphology and cell death.

chloramphenicol (Chlr) where indicated. Antibiotic susceptibility disks were purchased from Thermo Fisher Scientific (Oxoid). Bacteria stocks were stored in 15% glycerol at -20˚C for short periods and -80˚C for long periods.

**Table 1. Plasmids and host bacterial strains.**

| Plasmid | Relevant features | *E. coli* host strains | Reference |
|---|---|---|---|
| **pUC19** | *hok/sok⁻*, Amp^R | Top 10, SS996 | [16] |
| **pCCB1** | *hok/sok⁺*, Amp^R | Top 10, SS996 | [16] |
| **pIAU80** | *hok/sok⁻, ftsZ-yfp*, Amp^R | Top 10, SS996 | [25] |
| **pCCB2** | *hok/sok⁺, ftsZ-yfp*,Amp^R | Top 10, SS996 | [16] |
| **pHNZ** | *hok/sok⁻, ftsZ-antisense*, Chlr^R | Top 10, SS996 | [26] |
| **pCCB3** | *hok/sok⁺, ftsZ-antisense*, Chlr^R | Top 10, SS996 | [16] |

## Preparation of competent cells

Host *E. coli* cells were made competent chemically (using $CaCl_2$) for subsequent transformation with the plasmids indicated. A single colony of the host cell was grown in 5 ml LB broth for 2–3 hrs with vigorous shaking (≈250 rpm), then transferred to 100 ml of pre-warmed LB broth at 37°C. This was incubated at 37°C with vigorous shaking (250 rpm) to an optical density of 0.6 at 600nm (checked with spectrophotometer). Cells were chilled on ice for 15 mins and transferred into 50ml sterile centrifuge tubes, which were then centrifuged at 2000 x g for 10mins, with temperature pre-set at 0°C. The pellets were then suspended in 15 ml of sterile ice-cold 0.1 M $CaCl_2$ by gentle pipetting, cooled on ice for 15 mins and centrifuged again for 10 mins. They were again suspended in 4 ml 0.1 M $CaCl_2$ in 15% glycerol and 200 µl aliquots were stored at -80°C until used.

## Plasmid extraction

Overnight cultures of the bacteria cells containing the desired plasmid were grown in LB media containing selective antibiotics appropriate for each plasmid in a Stuart orbital incubator S1500 at 37°C for 16 hrs with shaking (180 rpm). Plasmids were extracted using QIAgene Miniprep kit (QIAGEN #27104), following the manufacturer's protocol.

## Transformation of bacteria with plasmids

Frozen competent bacteria were allowed to thaw gently on ice. 5 µl of plasmid (≈5 ng) was then added to 50 µl of competent cells in 1.5 ml microfuge tubes, mixed gently and incubated on ice for 30mins. The cells were shocked by placing the tube in a water bath at 42°C for 20–30 sec. Cells were recovered by incubating with 500 µl of SOC medium for 1–2 hrs, and then 50 µl were plated on LB agar plates containing appropriate antibiotics. After overnight incubation, a single colony was streaked on agar plate containing the selective antibiotic for further growth. To further check the success of the transformation, plasmid extraction was carried out with the transformants using QIAprep miniprep kit and the resulting plasmid band checked against the size of the original plasmid by electrophoresis on 1% agarose gel. Glycerol stocks of transformants were stored at -80°C.

## Growth and antibiotic susceptibility assays

Overnight bacterial cultures were diluted to about $1 \times 10^6$ CFU (approximately 1 in 1000 dilution) in 96 well plates and grown as already described [16]. The antibiotic susceptibility assays were done by both disk diffusion antibiotic testing and broth dilution method. MIC values were scored as the lowest concentration of the drug at which no observable growth of the bacterial culture occurred after 18hrs of incubation.

## Phase contrast microscopy

Samples (200 µl) were collected from bacterial cultures treated with sub-inhibitory concentration of doxycycline (0.5 µM) and processed as earlier described [27]. The number of cells showing the ghost cell morphology was counted per field of view, and the average expressed as percentages in relation to the normal cell morphology.

## Data analysis

Data was analyzed using Microsoft Office Excel 2017 and IBM SPSS Statistics version 21. Culture growth and morphology data were analyzed using descriptive statistics, while the mean doubling times and average growth rates of *hok/sok*+ and *hok/sok*- cultures were compared

using non-parametric paired samples test (Wilcoxon signed-rank test). The frequency of occurrence of ghost cells in *hok/sok*+ and *hok/sok*- cultures was compared using Chi-square test at 1 degree of freedom. Statistical significance was considered at *P*-value ≤ 0.05.

## Results

### Effect of the *hok/sok* locus on the susceptibility of *E. coli* (Top 10) to doxycycline

To assess the *hok/sok* locus effects in a stable background, we used *E. coli* Top 10 strain, in which both *hok/sok*+ and control plasmids, pCCB1 and pUC19 are very stable without antibiotic selection. Antibiotic susceptibility disk diffusion tests were done with doxycycline disks on LB agar plates. The results showed that the zone of inhibition is increased in cells containing the *hok/sok* locus when compared to the *hok/sok*- cells (Fig 2), indicating that the *hok/sok*+ cells are more susceptible to doxycycline. To confirm this observed increase in susceptibility of the *hok/sok*+ cells and determine the extent to which the *hok/sok* locus affects doxycycline susceptibility, antibiotic susceptibility assay was also carried out in broth media using 96 well plates. The results also showed that the *hok/sok*+ cells were more susceptible to doxycycline than the *hok/sok*- cells. Lower concentrations of doxycycline were needed to inhibit growth in cells containing the *hok/sok* locus than was required in cell lacking the locus (Fig 2). Whereas 0.6 μM doxycycline completely inhibited the growth of *hok/sok*+ cells, the *hok/sok*- still showed measurable growth beyond 0.7μM doxycycline concentration. Although the *hok/sok* locus confers resistance to certain antibiotics, our results show contrasting effects for doxycycline, where the locus confers increased doxycycline susceptibility.

### Effect of the *hok/sok* locus on doxycycline susceptibility of SOS-negative *E. coli* strain (SS996)

It has been established that the *hok/sok* locus acts as a stress response element in bacteria, and can functionally complement a defective SOS response [16]. Here, we explored the effect of the *hok/sok* locus on doxycycline susceptibility of an *E. coli* strain that is SOS-defective (SS996). This strain is ordinarily unable to withstand stressful growth conditions such as high temperature and antibiotic treatment, with very limited growth in such conditions. However, with the *hok/sok* locus, it grows unrestricted in such stressful growth conditions. Doxycycline susceptibility tests showed that the *hok/sok*+ cells were not only more susceptible, but the increase in susceptibility of the *hok/sok*+ cells to doxycycline is much more pronounced in this strain than in the Top 10 strain (Fig 3). While 1 μM doxycycline completely inhibited the growth of the *hok/sok*+ cells, the *hok/sok*- cells was not inhibited to even half the normal level at the same drug concentration. This is opposite to previous observations made when using other antibiotics such as ampicillin and amoxicillin, where the *hok/sok*- cells were more susceptible [16]. This suggests that the growth enhancing effects of the *hok/sok* locus also increases its susceptibility to doxycycline. In other words, the more the *hok/sok* locus enhances the bacterial stress response (and survivability in the absence of doxycycline), the more it increases the host cell susceptibility to doxycycline.

### Effect of doxycycline on the growth curve characteristics of hok/sok host *E. coli*

The *hok/sok* locus has been reported to significantly increase the lag time and rate of exponential growth phase of host bacteria cells [16]. When these *hok/sok* host cells were treated with increasing doses of doxycycline, there was a dose-dependent decrease in the average growth

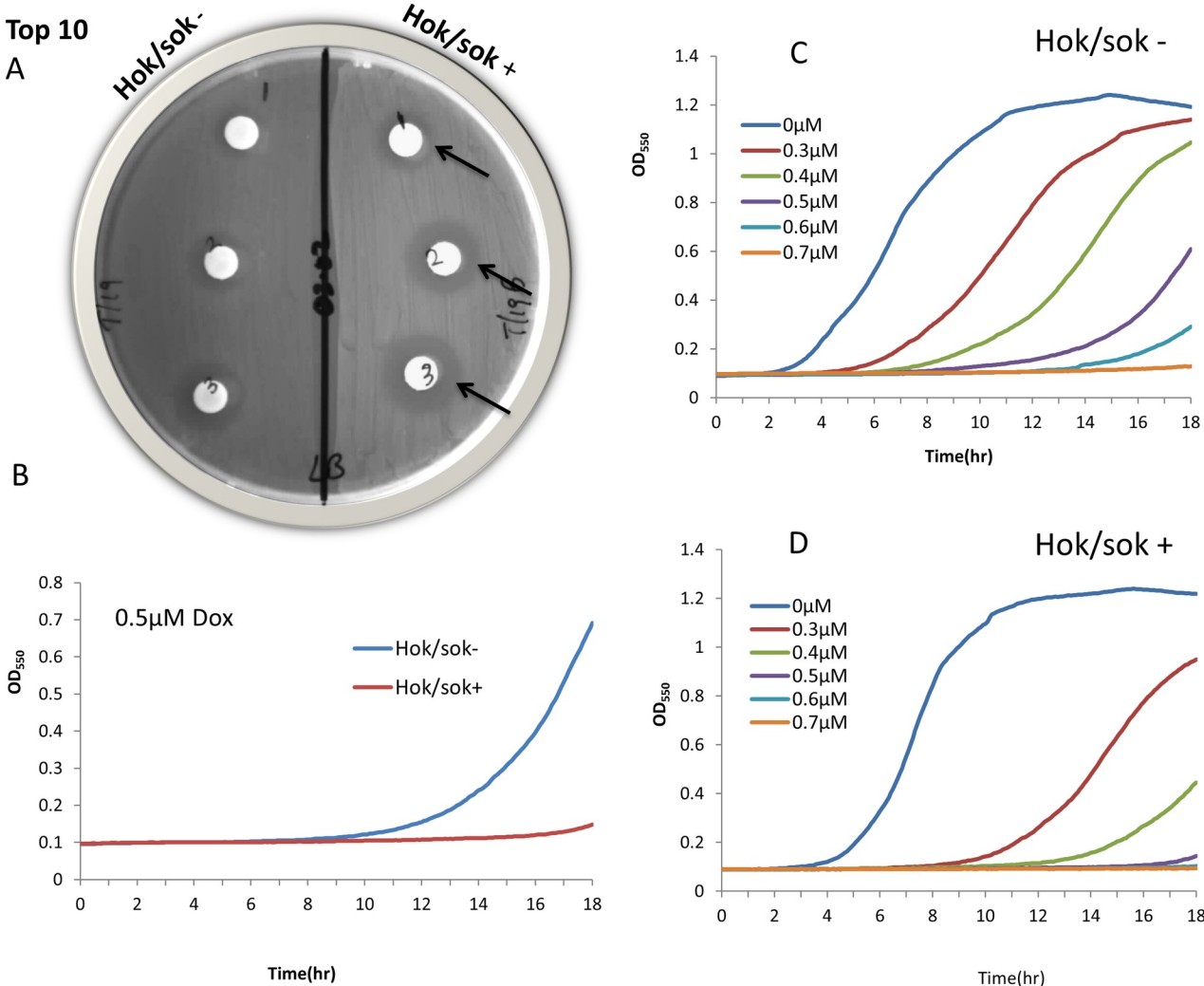

**Fig 2. Effect of the *hok/sok* locus on *E. coli* K-12 strain (Top 10) susceptibility to doxycycline.** Image of disk diffusion susceptibility of *E. coli* Top 10 strain. Increasing amounts of doxycycline were applied to the disks (1, 2, 3 = 0.1, 0.2, 0.3 μg doxycycline). Arrows indicate wider zones of inhibition in *hok/sok*+ cells compared to the control strain. Graphs show growth curves of cells in increasing concentrations of doxycycline, with the *hok/sok*+ cells showing higher growth inhibition. Data is representative of six repeat experiments.

rate of the *hok/sok*- Top 10 cells (as would be expected of any antibiotic). However, this decrease occurred in the *hok/sok*+ cells despite the *hok/sok*-induced increase in growth rate (Table 2), with significantly lower growth rates for the *hok/sok*+ cells than the *hok/sok*- cells ($p = 0.001$). This indicates that doxycycline inhibits culture growth in the *hok/sok*+ *via* an additional mechanism to its normal inhibitory mechanism in control cells. Furthermore, the *hok/sok* locus increased the doubling time of host cells (as expected). The difference between the doubling times of the *hok/sok*+ and *hok/sok*- cells was also significantly increased with doxycycline treatment ($p = 0.011$). This suggests that doxycycline may inhibit culture growth via *hok/sok*-induced growth effects. In the SS996 strain, doxycycline did not reduce the average growth rate of the control cells at lower doses (the growth rate was rather increased at doses ≤0.6 μM), indicating that the growth of the SS996 cells are not inhibited by the antibacterial activity of doxycycline at these doses. Conversely, doxycycline induced a significant decrease in the growth rate of the *hok/sok*+ SS996 cells ($p = 0.001$) in a dose-dependent manner, despite the

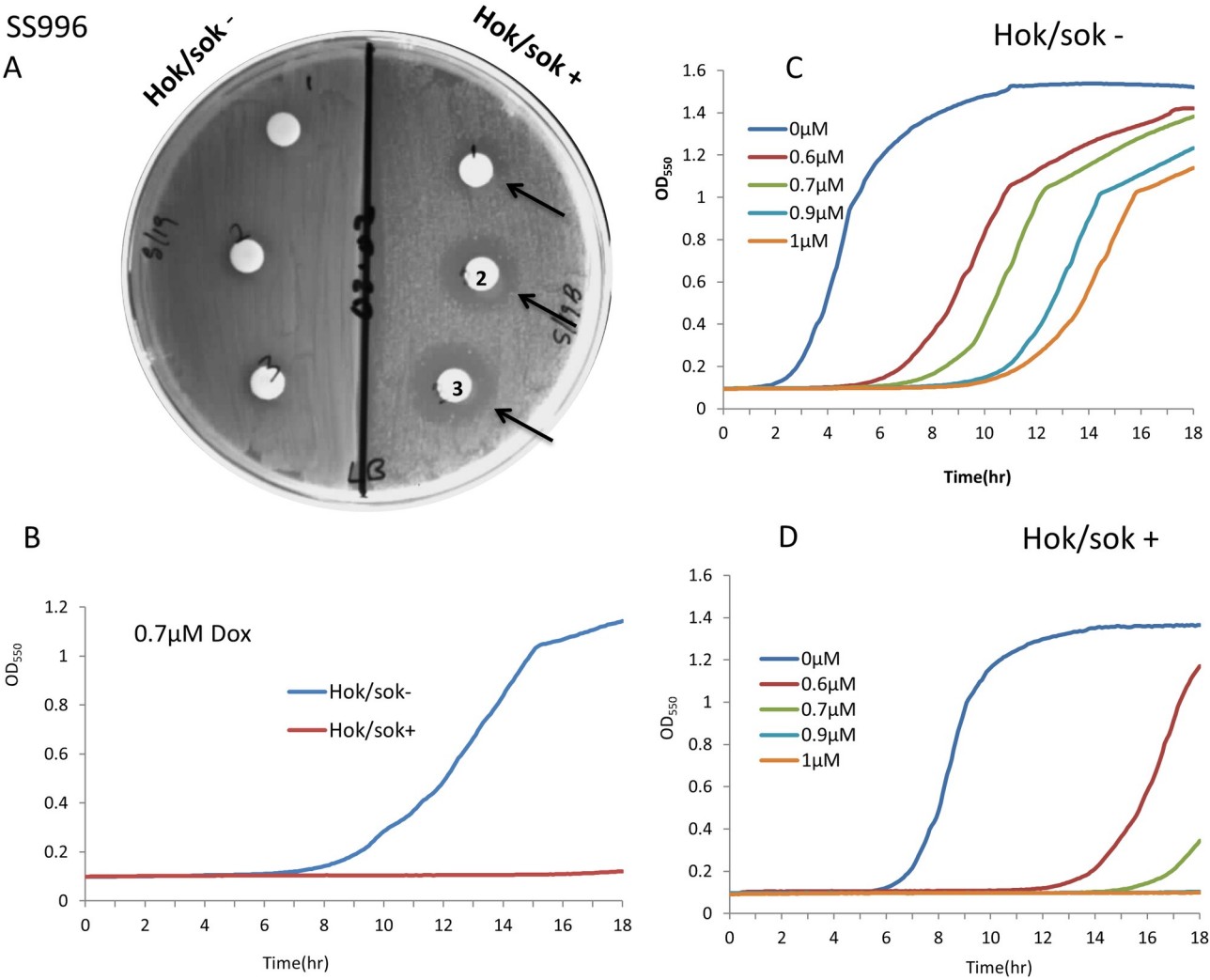

**Fig 3. Effect of the *hok/sok* locus on the susceptibility of SOS-deficient *E. coli* (SS996) to doxycycline.** Susceptibility of *E. coli* SOS-deficient strain (SS996) with increasing concentrations of doxycycline applied to disks (1, 2, 3 = 0.1, 0.2, 0.3μg doxycycline). Arrows indicate wider zones of inhibition in *hok*/sok⁺ cells compared to the control strain. Graphs show growth curves of cultures containing increasing concentrations of doxycycline, with the *hok/sok*⁺ cells showing higher growth inhibition. Data is representative of six repeat experiments.

higher growth rate of the untreated *hok/sok*⁺ cells when compared to the control cells. This indicates that the *hok/sok* locus could counteract the mechanism by which the SS996 persist at lower doxycycline concentrations, thus rendering the cells susceptible to doxycycline at all doses.

### Effect of the *hok/sok* associated β-lactam resistance on doxycycline susceptibility

The *hok/sok* locus has been shown to increase host cell tolerance/resistance to the β-lactam antibiotics (specifically ampicillin and amoxicillin), and this is more apparent in the SOS-deficient strain, SS996 [16]. Hence, we also explored the effect of *hok/sok*-associated β-lactam resistance on the susceptibility of the cells to doxycycline. In the Top 10 strain, the results showed that the *hok/sok*⁺ cells were still more susceptible to doxycycline than the *hok/sok*⁻ cells even in the presence of ampicillin (Fig 4). When the growth of the cells was compared in LB

**Table 2. Average growth rates (Avg GR) and doubling times of doxycycline-treated *E. coli*.**

| Dose of Doxy (µM) | Top 10 | | | | | SS996 | | | | |
|---|---|---|---|---|---|---|---|---|---|---|
| | Doubling Time (hh.mm) | | | Avg GR | Avg GR | Doubling Time (hh.mm) | | | Avg GR | Avg GR |
| | Hok/sok- (pUC19) | Hok/sok+ (pCCB1) | Diff | Hok/sok- (pUC19) | Hok/sok+ (pCCB1) | Hok/sok- (pUC19) | Hok/sok+ (pCCB1) | Diff | Hok/sok- (pUC19) | Hok/sok+ (pCCB1) |
| 0 | 03:40 | 04:55 | 01:15 | 1.954 | 2.094 | 02:45 | 06:50 | 04:05 | 2.183 | 2.437 |
| 0.2 | 05:05 | 08:05 | 03:00 | 1.942 | 1.732 | 03:30 | 08:20 | 04:50 | 2.256 | 2.354 |
| 0.3 | 06:40 | 10:45 | 04:05 | 1.785 | 1.078 | 03:55 | 09:30 | 05:35 | 2.308 | 2.269 |
| 0.4 | 09:00 | 14:25 | 05:25 | 1.271 | 0.297 | 05:05 | 10:20 | 05:15 | 2.473 | 2.198 |
| 0.5 | 13:05 | >18:00 | >5:00 | 0.445 | 0.037 | 05:50 | 12:30 | 06:40 | 2.448 | 1.382 |
| 0.6 | 16:05 | >18:00 | - | 0.156 | 0.013 | 06:45 | 13:45 | 07:00 | 2.332 | 0.833 |
| 0.7 | >18:00 | >18:00 | - | 0.033 | 0.005 | 08:20 | 16:15 | 07:55 | 2.157 | 0.207 |
| 0.8 | >18:00 | >18:00 | - | 0.019 | 0.003 | 09:55 | 16:35 | 06:40 | 1.779 | 0.120 |
| 0.9 | >18:00 | >18:00 | - | 0.014 | 0.001 | 10:45 | >18:00 | >7:15 | 1.655 | 0.004 |
| 1.0 | >18:00 | >18:00 | - | 0.007 | 0.001 | 11:15 | >18:00 | >7:00 | 1.358 | 0.007 |

media with and without ampicillin, there was no difference in the growth pattern and doxycycline susceptibility of the cells (D). This indicates that doxycycline susceptibility of the cells is not affected by the presence of ampicillin in the media (and the associated resistance), suggesting that the increase in doxycycline susceptibility in *hok/sok*+ cells occurs via a mechanism that is not antagonistic to the mechanism of ampicillin resistance.

In the SS996 strain, there was also no difference in the growth pattern and doxycycline susceptibility of the *hok/sok*+ cells when compared in LB and LB amp media (Fig 5). This further indicates that doxycycline susceptibility of the *hok/sok*+ cells is not affected by the associated β-lactam (ampicillin) resistance. However, the control (*hok/sok*-) cells showed greater growth inhibition and improved susceptibility to doxycycline in LB amp media, due to its increased susceptibility to ampicillin [16]. These results indicate that doxycycline can overcome the ampicillin resistance imparted by the *hok/sok* locus to host cells, and that both doxycycline and ampicillin can work synergistically to enhance bacterial killing/growth inhibition.

## MIC of doxycycline in *hok/sok* host cells

MIC determination for doxycycline also showed that lower amounts of doxycycline are required to completely inhibit the growth of cells containing the *hok/sok* than those that do not contain the *hok/sok* locus (Table 3).

## Effect of doxycycline on the growth of *ftsZ*-mutant *hok/sok* plasmid host cells

To further test that the observed increase in doxycycline susceptibility is mediated by the *hok/sok* locus, we also tested the susceptibility of cells containing two other *hok/sok* plasmids (pCCB2 and pCCB3) and their control cells (plAU80 and pHNZ) to doxycycline. The results still showed increased susceptibility of the *hok/sok*+ cells to doxycycline (Figs 6 and 7). This further asserts that the *hok/sok* locus enhances host cell susceptibility to doxycycline.

## Effects of doxycycline on the morphology of *hok/sok* host cells

The *hok/sok* toxin/antitoxin system induces host cell death *via* a post-segregational killing mechanism that is associated with a characteristic cell morphology known as "ghost cells",

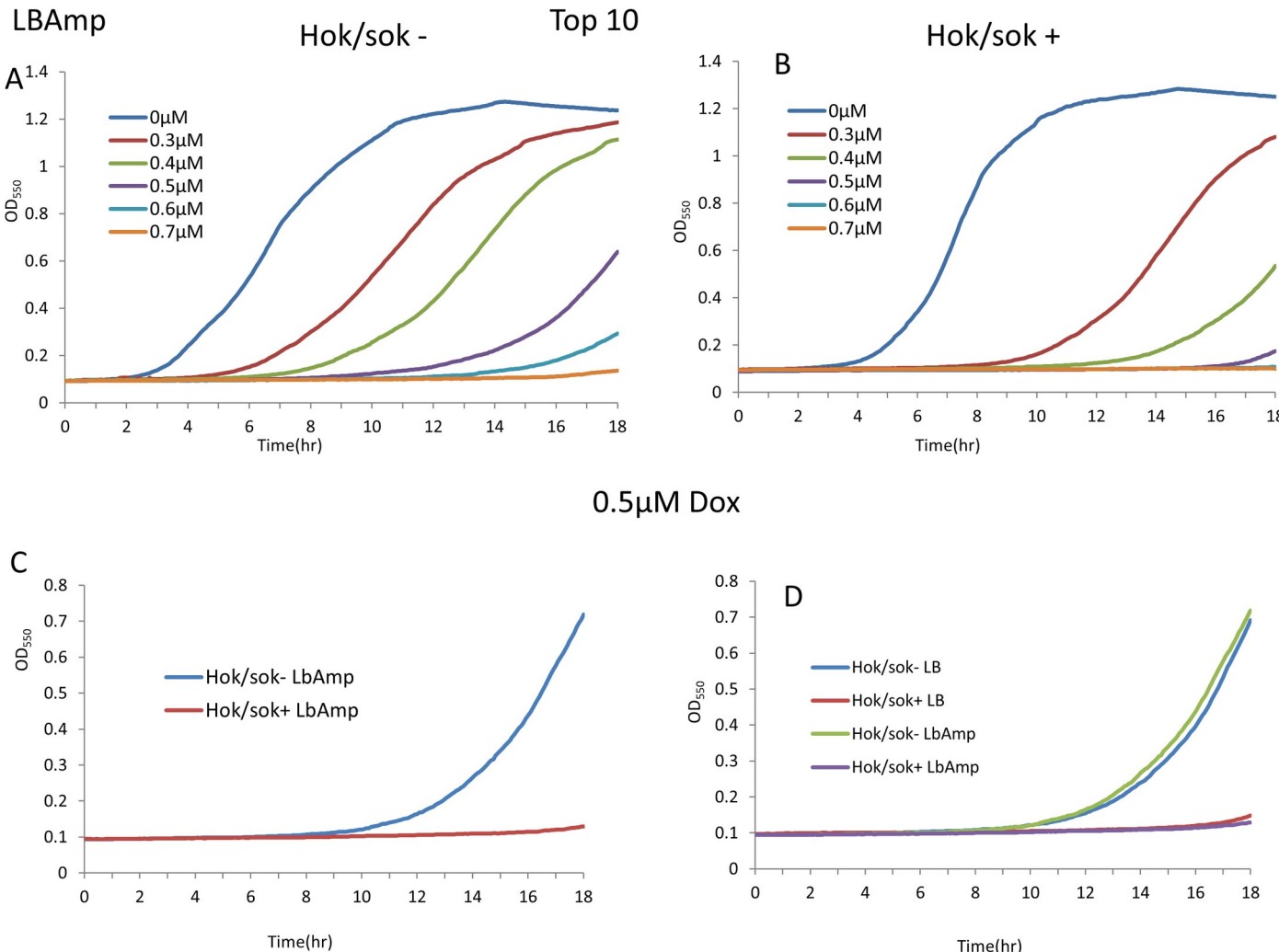

**Fig 4. Effect of the hok/sok associated β-lactam resistance on doxycycline susceptibility.** Graphs show growth curves of Top 10 cells in increasing concentrations of doxycycline in LB amp media (containing 100 μg/ml ampicillin), with the *hok/sok*+ cells showing higher growth inhibition (C). Doxycycline susceptibility of the *hok/sok*+ cells remains the same in both LB and LB amp media (D). Data is representative of six repeat experiments.

which arise due to the activity of the Hok toxin. To ascertain whether the observed increase in doxycycline susceptibility of *hok/sok* host cells is connected to the toxin/antitoxin system, we examined the morphology of cells treated with sub-inhibitory concentration of doxycycline (0.5 μM ≈0.2 μg/ml) by phase contrast microscopy to see whether the cells show the typical morphology of Hok toxin activity/cell death in bacteria (ghost cells). The images obtained show significantly increased number (*p* = 0.000) of the *hok/sok*+ cells with morphologic changes suggestive of Hok toxin activity (ghost cells). These cells appear denser at the cell poles, indicating impaired membrane permeability associated with Hok killing (Fig 8). About 70% of the *hok/sok*+ cells showed ghost cell morphology, whereas less than 10% of the *hok/sok*- cells showed a similar morphology (similar to the untreated cells). This suggests that the Hok toxin is activated in bacteria cells containing the *hok/sok* locus following doxycycline treatment.

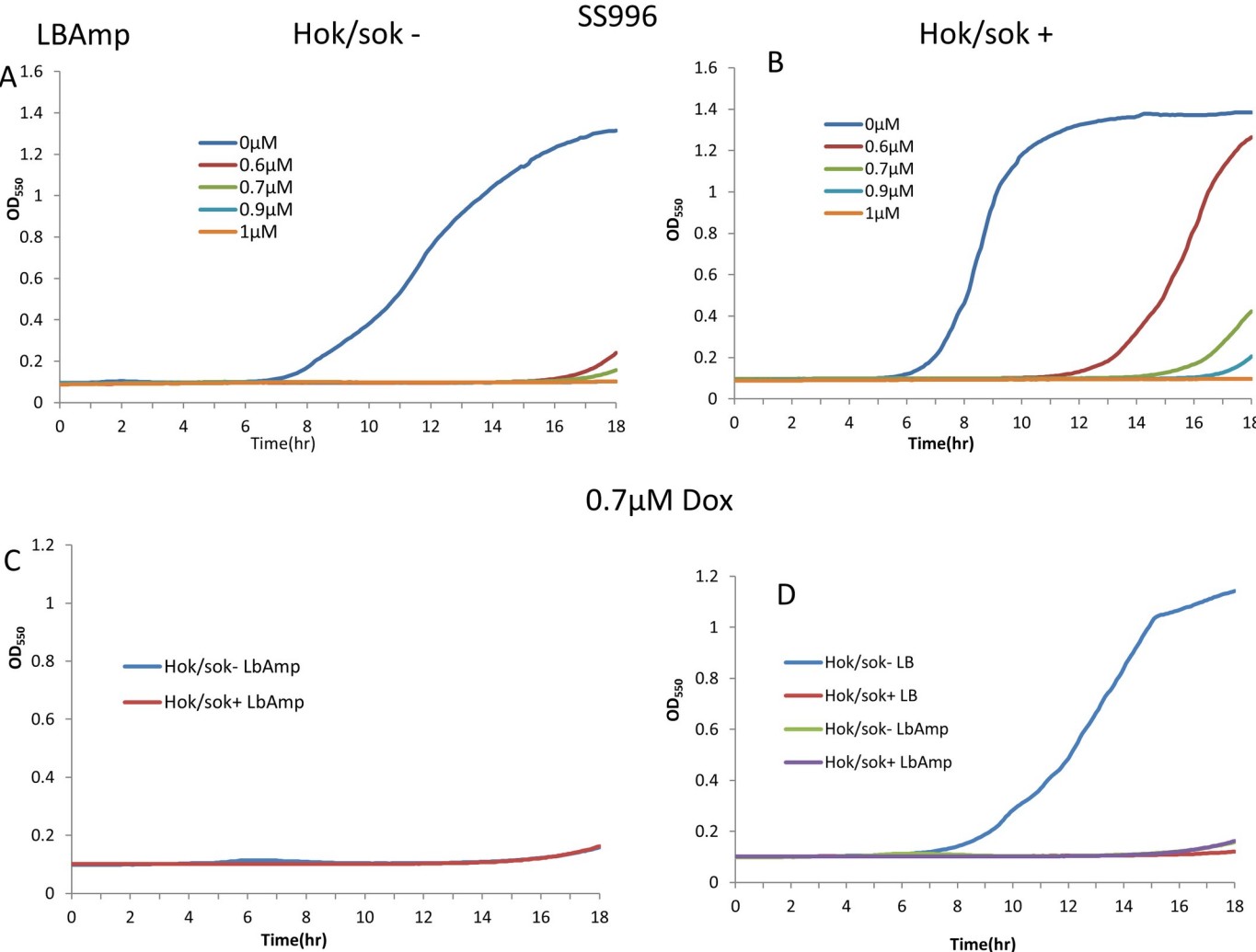

**Fig 5. Effect of the hok/sok associated β-lactam resistance on doxycycline susceptibility.** Graphs show growth curves of SS996 cells in increasing concentrations of doxycycline in LB amp media (containing 100 μg/ml ampicillin). Doxycycline susceptibility of the *hok/sok*+ cells remains the same in both LB and LB amp media (D), but the *hok/sok*- cells also show increased susceptibility in LB amp media. Data is representative of six repeat experiments.

## Discussion

The *hok/sok* locus can contribute to bacterial stress responses, thereby helping cells survive adverse growth condition such as high temperature and antibiotic treatment [16]. The locus increases both the lag time and exponential growth rates of host bacteria cultures. In this study, we show that whereas the *hok/sok* will generally increase doubling time and average growth rate of host bacteria cultures, it inadvertently confers increased doxycycline

**Table 3. MIC values (μg/ml) of doxycycline in *E. coli* cells containing the plasmid-borne *hok/sok* locus.**

| Host strain | Media | *Hok/sok*- (pUC19) | *Hok/sok*+ (pCCB1) |
|---|---|---|---|
| **Top 10** | LB | 0.34 | 0.24 |
| **Top 10** | LB amp | 0.34 | 0.24 |
| **SS996** | LB | 0.80 | 0.38 |
| **SS996** | LB amp | 0.38 | 0.38 |

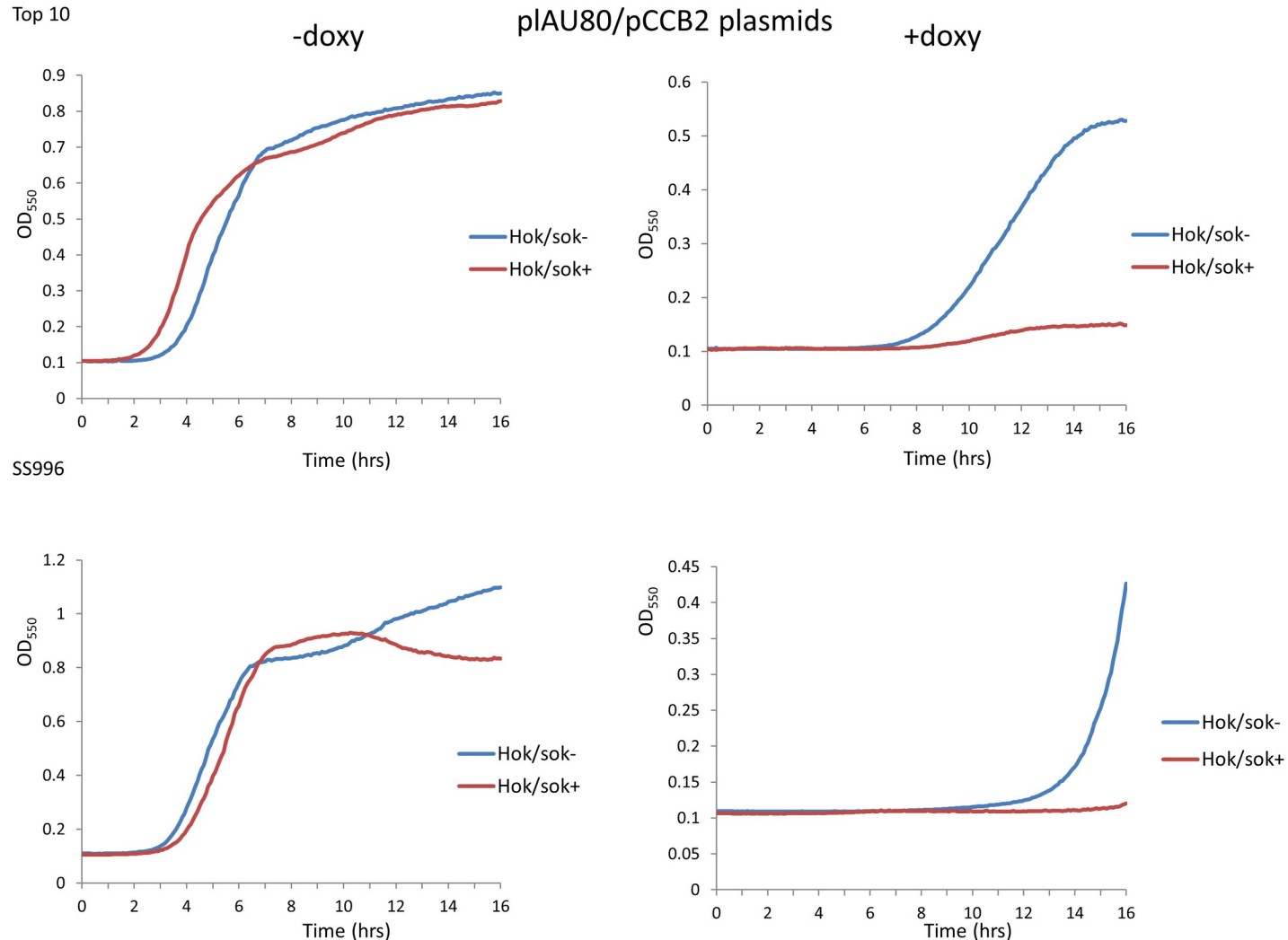

**Fig 6. Effect of the *hok/sok* locus on doxycycline susceptibility.** Growth curves of Top 10 and SS996 cells containing pCCB2 *hok/sok* plasmids and controls treated with 1 µM doxycycline, with the *hok/sok*+ cells showing increased susceptibility.

susceptibility to host cells especially at very low doses that the cells would otherwise not have been susceptible. This increased susceptibility to doxycycline appear to be mediated via an additional mechanism that acts synergistically with its established antibacterial mechanism of protein synthesis inhibition. This is particularly interesting, considering the fact that the *hok/sok* locus enhances ampicillin (and β-lactam antibiotics) resistance in host bacteria cells [16].

The observed increase in doxycycline susceptibility in this study was found to be more prominent in the SOS-deficient strain (SS996) than in the normal strain (Top 10). This is consistent with previous reports that the growth effects of the *hok/sok* locus is more prominent in strains that are defective in the SOS response [16], and has been associated with alterations in FtsZ activity [27]. The SOS response pathway has been implicated in bacterial persistence to β-lactam antibiotics and the fluoroquinolones [28, 29], but conversely increases doxycycline susceptibility in this study.

The *hok/sok* locus is closely associated with plasmids encoding ESBLs, especially CTX-M beta-lactamases [20], and many other antibiotics resistance genes [17–19]. It contributes to β-

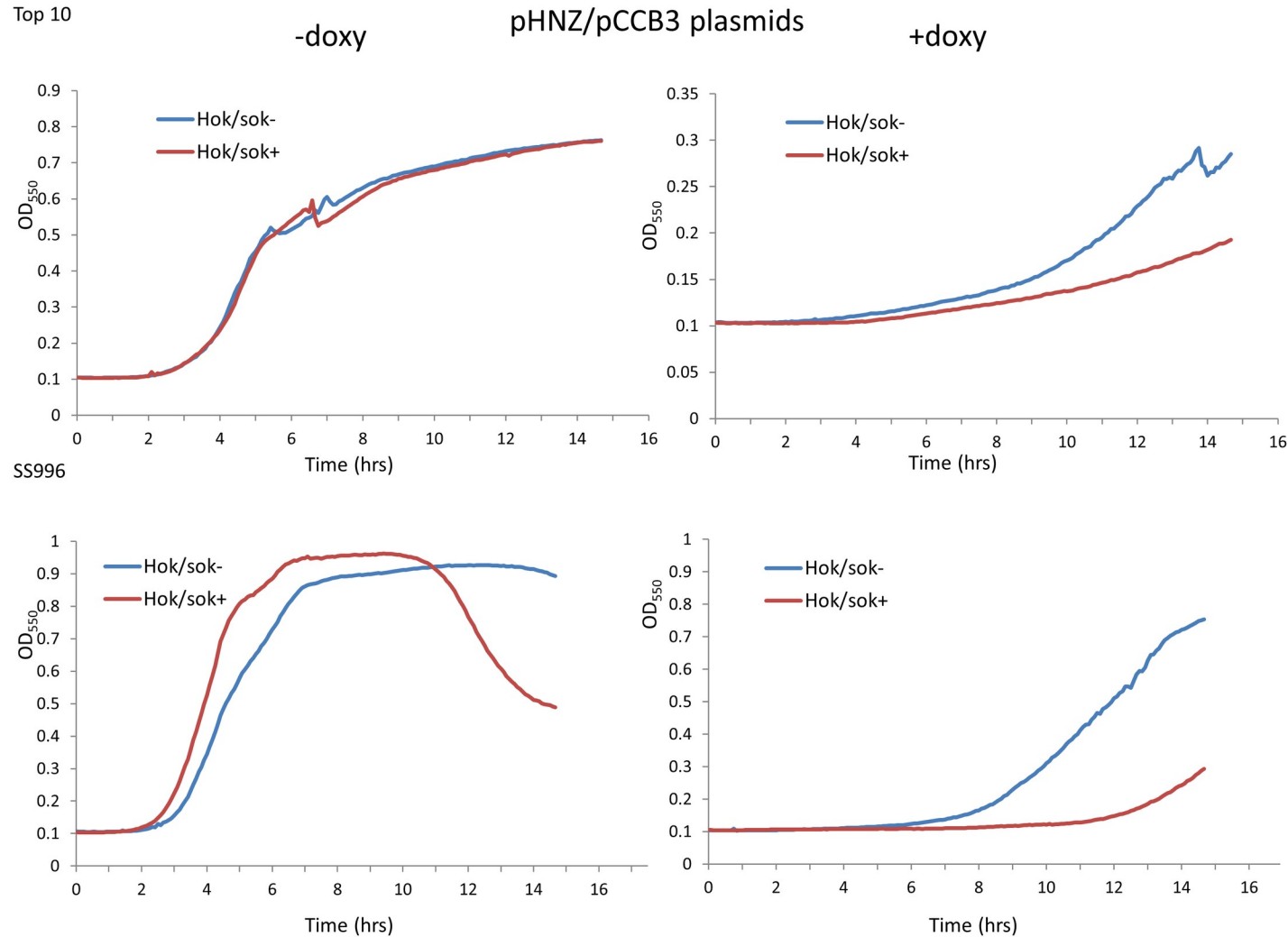

**Fig 7. Effects of the *hok/sok* locus on doxycycline susceptibility.** Growth curves of Top 10 and SS996 cells containing pCCB3 *hok/sok* plasmids and controls treated with 1 μM doxycycline, with the *hok/sok⁺* cells showing increased susceptibility.

lactam resistance in two ways: by increasing the tolerance of host cells to higher amounts of the drug [16], and by ensuring the stabilization/maintenance and propagation of the resistance genes normally carried on the *hok/sok* plasmids [15, 30]. The plasmid stabilization/maintenance function of the *hok/sok* locus confers traits that help bacteria to evade multiple antibiotic treatment, thus sustaining and spreading antibiotic resistance. The combination of multiple mechanisms of enhancing antibiotic resistance makes understanding of the *hok/sok* gene critical in the fight against antibiotic resistance. Hence, the observations of increased *hok/sok* host cell susceptibility to doxycycline in this study could open up opportunities to fight multiple antibiotic resistance mechanisms using the bacterial intrinsic survival mechanisms. It also gets more interesting to observe that doxycycline, a member of the tetracycline group of antibiotics whose clinical usefulness has been greatly limited due to the development of resistance, could be useful in this regard.

When the doxycycline susceptibility of the *hok/sok* host cells was compared in media with and without ampicillin, there was no difference in the growth pattern of the cells. This indicates that the *hok/sok* locus renders bacteria more susceptible to doxycycline irrespective of

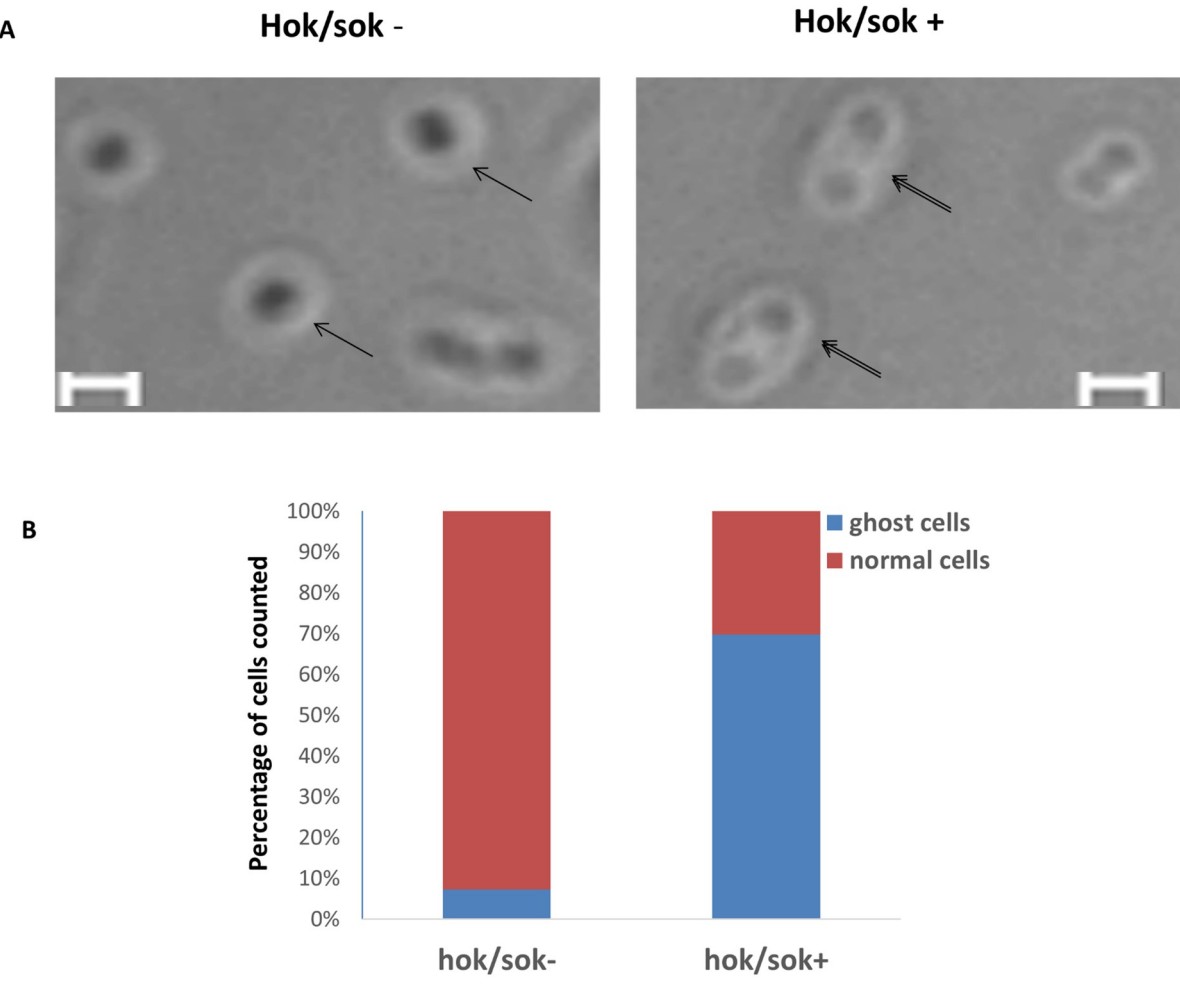

**Fig 8. Phase contrast microscopy images of Top 10 *hok/sok*⁺ and *hok/sok*⁻ cells treated with sub-inhibitory concentration of doxycycline (0.5 μM).** Double arrows indicate cells that appear denser at the poles than the middle ("ghost" cells), which are morphological changes associated with Hok toxin activity, compared to normal cells (indicated by single arrow). Scale bar = 2 μm. Histogram shows the percentage of ghost cells to normal cells counted from an equal area of the phase contrast microscopy images of *hok/sok*⁺ and *hok/sok*⁻ bacteria samples, with about 70% ghost cells in the *hok/sok*⁺ sample and about 10% ghost cells in the *hok/sok*⁻ sample.

resistance to other antibiotics, suggesting that the *hok/sok*-mediated increase in doxycycline susceptibility occurs via a mechanism that is not antagonistic to the mechanism of ampicillin resistance. Hence, doxycycline may provide therapeutic advantage against infections by pathogens containing the *hok/sok* locus, particularly in cases of suspected beta-lactam resistance. Again, the synergistic action of doxycycline with ampicillin may present a better therapeutic combination in cases where the genetic makeup of the cells cannot be determined.

The observed morphologic appearance of the *hok/sok* host cells treated with doxycycline suggests that *hok/sok* locus enhances doxycycline susceptibility by inducing Hok toxin expression and killing of the cells. Since doxycycline inhibits RNase III cleavage and processing of dsRNAs [23, 24], it could also inhibit RNase III degradation of the *hok* mRNA:Sok RNA complex leading to the eventual decay of the labile Sok RNA and releasing the *hok* mRNA for translation and subsequent cell killing. Activation of Hok toxin activity/self-killing in bacteria containing *hok/sok* plasmids has also been reported by Faridani et al (2006) using peptide

nucleic acids (PNA) that target Sok RNA [22]. This suggests that antimicrobial agents that could inhibit RNase degradation of the *hok*/*sok* dsRNA complex (such as RNA ligands) could therefore be exploited as a control and therapeutic strategy to reduce multi-drug resistance plasmid dissemination and induce self-killing in the cells, thereby enhancing antibiotic efficacy as exemplified here with doxycycline. In future, it will be interesting to see how these strategies perform in clinical isolates.

## Author Contributions

**Conceptualization:** Chinwe Uzoma Chukwudi, Liam Good.

**Data curation:** Chinwe Uzoma Chukwudi, Liam Good.

**Formal analysis:** Chinwe Uzoma Chukwudi.

**Funding acquisition:** Chinwe Uzoma Chukwudi.

**Investigation:** Chinwe Uzoma Chukwudi.

**Methodology:** Chinwe Uzoma Chukwudi, Liam Good.

**Project administration:** Liam Good.

**Supervision:** Liam Good.

**Validation:** Chinwe Uzoma Chukwudi, Liam Good.

**Visualization:** Liam Good.

**Writing – original draft:** Chinwe Uzoma Chukwudi.

**Writing – review & editing:** Chinwe Uzoma Chukwudi, Liam Good.

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
