## [Decision Letter · Decision Letter 0]

23 Apr 2020

PONE-D-20-06337

DOXYCYCLINE INDUCES HOK TOXIN KILLING IN HOST E. COLI

PLOS ONE

Dear Dr Chukwudi,

Thank you for submitting your manuscript to PLOS ONE. After careful consideration, we feel that it has merit but does not fully meet PLOS ONE’s publication criteria as it currently stands. Therefore, we invite you to submit a revised version of the manuscript that addresses the points raised during the review process.

We would appreciate receiving your revised manuscript by Jun 07 2020 11:59PM. To enhance the reproducibility of your results, we recommend that if applicable you deposit your laboratory protocols in protocols.io, where a protocol can be assigned its own identifier (DOI) such that it can be cited independently in the future. For instructions see: http://journals.plos.org/plosone/s/submission-guidelines#loc-laboratory-protocols

We look forward to receiving your revised manuscript.

Kind regards,

Monica Cartelle Gestal, PhD

Academic Editor

PLOS ONE

Journal Requirements:

2. Please include a copy of Table 2 which you refer to in your text on page 11.

Reviewers' comments:

Reviewer's Responses to Questions

**Comments to the Author**

1. Is the manuscript technically sound, and do the data support the conclusions?

Reviewer #1: Yes

Reviewer #2: Partly

2. Has the statistical analysis been performed appropriately and rigorously? 

Reviewer #1: Yes

Reviewer #2: No

3. Have the authors made all data underlying the findings in their manuscript fully available?

Reviewer #1: Yes

Reviewer #2: Yes

4. Is the manuscript presented in an intelligible fashion and written in standard English?

Reviewer #1: Yes

Reviewer #2: Yes

5. Review Comments to the Author

Reviewer #1: Interesting article, suggested changes are minor

Please review the attached file

It is suggested to check the magazine regulations regarding the number of figures that appear in an article.

In general, it is recommended to improve the quality of the graphics and in particular figure 8, which is presented with low quality and makes it difficult to appreciate the characteristics of the ghost cells

Reviewer #2: The manuscript provides preliminary evidence supporting the hypothesis that doxycycline inhibition of the bacterial hok/sok dsRNA complex degradation by RNAse III consequently enhances the killing effect of hok. Furthermore, this effect of doxycycline appears independent of other antibiotics and could be exploited to provide synergistic and more effective outcomes in treatment of antibiotic resistance mediated via ESBL-gene harboring plasmids.

The work has a logical hypothesis founded on substantial preexisting work in the field, and the manuscript reads reasonably well. However, three major caveats detract from the manuscript. Firstly, the relevant results involve SOS deficient E.coli lab strains where the effects are most pronounced, but how these extrapolate to wild type is not discussed; and secondly, it is assumed all along that all observed growth effects are caused in a dose dependent manner by the targeted action of doxycycline on RNAse III as opposed to other non-targeted effects. Third, the manuscript lacks any statistical inferences.

I recommend that the manuscript be re-written resubmitted to include the discussion addressing these major caveats.

There are other instances where changes are required

• Presenting and tabulating differences in growth in a quantitative manner (rate of growth) with significance values

• Line:140

It is not clear what experiment the section starting at Line 140 refers to or, what “both plasmids” in line 146 relate too.

• Line 182-186

The description here is unnecessarily long and needs to be rephrased for clarity. Figure 3 shows the dose dependent inhibition of a “rescue” phenotype in hok/sok+ SOS- bacteria. It is assumed in the manuscript that this is a direct result of the doxycycline inhibition of RNASe III but is not proven. This has to be clearly indicated in a discussion and mentioning alternative explanations along with arguments against them.

Some minor points

• 178: Suggest starting with “While 1uM completely inhibited hok/sok+……”

• 170: Sentence in line 170 needs references.

• 172: The name of the SOS strain needs to indicated here

• 202: In the Top 10 strain,

• 205: Rewrite to “….compared in LB media with and without ampicillin”

6. PLOS authors have the option to publish the peer review history of their article (what does this mean?). If published, this will include your full peer review and any attached files.

Reviewer #1: No

Reviewer #2: No

---

## [Author Response · Author response to Decision Letter 0]

2 Jun 2020

DOXYCYCLINE 1 INDUCES HOK TOXIN KILLING IN HOST E. COLI

Reviewer's recommendations

Dear authors

In general it is an interesting article and the changes are minor

Line 100 correct write CaCl2-Cacl corrected to CaCl

Line 142 to line 147, all this information has already been presented in the introduction and in the methodology, so it is recommended to go to the results directly-lines 142-145 deleted. Opening sentence modified.

Line 216 the name of the medium used capitalize it LB and LB amp-done

Line 231 and 232 the names of the media used must be capitalized LB LB amp- done

Table 1. The names of the media used must be capitalized LB and LB amp- done

It is recommended to improve the presentation of figure 1 and avoid that the arrows are above the text-both presentation and quality improved.

Figure 8 is pixelated, improve the quality of it-Fig 8 quality improved.

Reviewer #2: The manuscript provides preliminary evidence supporting the hypothesis that doxycycline inhibition of the bacterial hok/sok dsRNA complex degradation by RNAse III consequently enhances the killing effect of hok. Furthermore, this effect of doxycycline appears independent of other antibiotics and could be exploited to provide synergistic and more effective outcomes in treatment of antibiotic resistance mediated via ESBL-gene harboring plasmids.

The work has a logical hypothesis founded on substantial preexisting work in the field, and the manuscript reads reasonably well. However, three major caveats detract from the manuscript. Firstly, the relevant results involve SOS deficient E.coli lab strains where the effects are most pronounced, but how these extrapolate to wild type is not discussed; The SOS deficient strain (SS996) was used to compare the effects observed in the normal lab strain (Top 10) because the hok/sok locus is known to function as a stress response element in E. coli , and complements bacterial SOS response. The discussion has been modified to more clearly explain the rational for using this strain. The following text has been added: “The observed increase in doxycycline susceptibility in this study was found to be more prominent in the SOS-deficient strain (SS996) than in the normal strain (Top 10). This is consistent with previous reports that the growth effects of the hok/sok locus is more prominent in strains that are defective in the SOS response (16), and has been associated with alterations in FtsZ activity (27). The SOS response pathway has been implicated in bacterial persistence to β-lactam antibiotics and the fluoroquinolones, but conversely increases doxycycline susceptibility in this study.” Also, with the discussion section, we explain that the effects will be examined in a range of clinical isolates in future.

and secondly, it is assumed all along that all observed growth effects are caused in a dose dependent manner by the targeted action of doxycycline on RNAse III as opposed to other non-targeted effects. This is a helpful comment. The inhibitory effects of antibiotics are often complex, and the growth effects observed in this study reflect both inhibition of RNAase III as well as other targets of tetracycline. We have extensively modified the manuscript with the addition of a section of analysing the growth effects of doxycycline in the light of other known inhibitory mechanisms. Here, we have demonstrated increased doxycycline susceptibility in the presence of the hok/sok locus, even though there is some level of susceptibility without the hok/sok (as would be expected of any antibiotics). To explain the situation more clearly, we have inserted additional text within the discussion section to explain that multiple antimicrobial mechanisms are possible, thus: “The locus increases both the lag time and exponential growth rates of host bacteria cultures. In this study, we show that whereas the hok/sok will generally increase doubling time and average growth rate of host bacteria cultures, it inadvertently confers increased doxycycline susceptibility to host cells especially at very low doses that the cells would otherwise not have been susceptible. This increased susceptibility to doxycycline appear to be mediated via an additional mechanism that acts synergistically with its established antibacterial mechanism of protein synthesis inhibition. This is particularly interesting, considering the fact that the hok/sok locus enhances ampicillin (and β-lactam antibiotics) resistance in host bacteria cells (16).”

Third, the manuscript lacks any statistical inferences. We have added inferential statistical analyses in the Data Analysis subsection of Materials and Methods. In the result section, we have added a new segment on “Effect of doxycycline on the growth curve characteristics of hok/sok host E. coli.” with p-values, which was also added to the segment on “Effects of doxycycline on the morphology of hok/sok host cells”.

I recommend that the manuscript be re-written resubmitted to include the discussion addressing these major caveats.

There are other instances where changes are required

• Presenting and tabulating differences in growth in a quantitative manner (rate of growth) with significance values. The rate of growth has now been calculated and presented clearly in table 2, and significance values included in the text..

• Line:140

It is not clear what experiment the section starting at Line 140 refers to or, what “both plasmids” in line 146 relate too.- the plasmid identities have been clarified: the text now reads “…. E. coli Top 10 strain, in which both hok/sok+ and control plasmids, pCCB1 and pUC19 are very stable without antibiotic selection)….”

• Line 182-186

The description here is unnecessarily long and needs to be rephrased for clarity. –Rephrased to now read “This suggests that the growth enhancing effects of the hok/sok locus also increases its susceptibility to doxycycline. In other words, the more the hok/sok locus enhances bacterial stress response and survivability, the more it increases the host cell susceptibility to doxycycline.”

Figure 3 shows the dose dependent inhibition of a “rescue” phenotype in hok/sok+ SOS- bacteria. It is assumed in the manuscript that this is a direct result of the doxycycline inhibition of RNASe III but is not proven. This has to be clearly indicated in a discussion and mentioning alternative explanations along with arguments against them. The results and discussion sections have been modified to include these considerations, as indicated above.

Some minor points

• 178: Suggest starting with “While 1uM completely inhibited hok/sok+……”-sentence modified, now reads “While 1 µM doxycycline completely inhibited the growth of the hok/sok+ cells, the hok/sok- cells was not inhibited to even half the normal level at the same drug concentration”

• 170: Sentence in line 170 needs references. Reference inserted

• 172: The name of the SOS strain needs to indicated here-indicated (SS996).

• 202: In the Top 10 strain,-done

• 205: Rewrite to “….compared in LB media with and without ampicillin”-done.

Journal Requirements:

1. Please ensure that your manuscript meets PLOS ONE's style requirements, including those for file naming. Checked and OK.

2. Please include a copy of Table 2 which you refer to in your text on page 11.-Table included (changed to Table 3).

---

## [Decision Letter · Decision Letter 1]

19 Jun 2020

DOXYCYCLINE INDUCES HOK TOXIN KILLING IN HOST E. COLI

PONE-D-20-06337R1

Dear Dr. Chukwudi,

We’re pleased to inform you that your manuscript has been judged scientifically suitable for publication and will be formally accepted for publication once it meets all outstanding technical requirements.

Within one week, you’ll receive an e-mail detailing the required amendments if needed. When these have been addressed, you’ll receive a formal acceptance letter and your manuscript will be scheduled for publication.

Kind regards,

Monica Cartelle Gestal, PhD

Academic Editor

PLOS ONE

Additional Editor Comments (optional):

Reviewers' comments:

Reviewer's Responses to Questions

**Comments to the Author**

1. If the authors have adequately addressed your comments raised in a previous round of review and you feel that this manuscript is now acceptable for publication, you may indicate that here to bypass the “Comments to the Author” section, enter your conflict of interest statement in the “Confidential to Editor” section, and submit your "Accept" recommendation.

Reviewer #1: All comments have been addressed

Reviewer #2: (No Response)

2. Is the manuscript technically sound, and do the data support the conclusions?

Reviewer #1: Yes

Reviewer #2: Yes

3. Has the statistical analysis been performed appropriately and rigorously? 

Reviewer #1: Yes

Reviewer #2: Yes

4. Have the authors made all data underlying the findings in their manuscript fully available?

Reviewer #1: Yes

Reviewer #2: Yes

5. Is the manuscript presented in an intelligible fashion and written in standard English?

Reviewer #1: Yes

Reviewer #2: Yes

6. Review Comments to the Author

Reviewer #1: Each of the recommendations made was subject to change and improvement, so that this version, already revised and corrected, is considered suitable for the respective publication, respecting the regulations of the journal for this.

Congratulations to the authors for their work

Reviewer #2: (No Response)

7. PLOS authors have the option to publish the peer review history of their article (what does this mean?). If published, this will include your full peer review and any attached files.

Reviewer #1: No

Reviewer #2: No

---

## [Editor Report · Acceptance letter]

24 Jun 2020

PONE-D-20-06337R1 

Doxycycline induces Hok toxin killing in host <I>E. coli</I> 

Dear Dr. Chukwudi:

I'm pleased to inform you that your manuscript has been deemed suitable for publication in PLOS ONE. Congratulations! Your manuscript is now with our production department. 

Kind regards, 

on behalf of

Dr. Monica Cartelle Gestal 

Academic Editor

PLOS ONE